# Screening and Identification of Drought-Tolerant Genes in Tomato (*Solanum lycopersicum* L.) Based on RNA-Seq Analysis

**DOI:** 10.3390/plants14101471

**Published:** 2025-05-14

**Authors:** Yue Ma, Yushan Li, Fan Wang, Quan Qing, Chengzhu Deng, Hao Wang, Yu Song

**Affiliations:** 1Institute of Crop Sciences, Xinjiang Academy of Agricultural Sciences, Urumqi 830091, China; my0988221@163.com (Y.M.); lys8302@163.com (Y.L.); 13999564298@163.com (F.W.); 2College of Horticulture, Xinjiang Agricultural University, Urumqi 830052, China; 17699527950@163.com (Q.Q.); 17842471207@163.com (C.D.); 3Institute of Fruits and Vegetables, Xinjiang Academy of Agricultural Sciences, Urumqi 830091, China

**Keywords:** tomato, drought resistance, RNA-seq, hub genes, VIGS

## Abstract

Drought is one of the major abiotic stresses that inhibits plant growth and development. Therefore, it is critical to explore drought resistance genes in crops to obtain high-quality breeding materials. In this study, the drought-sensitive tomato line “FQ118” and the resistant line “FQ119” were treated with PEG-6000 and, at 0 h (CK), 6 h, 24 h, 36 h and 48 h, the plants were evaluated for growth and physiological indicators, and leaf tissues were collected for RNA-seq. The growth indicators (growth trend, dry and fresh weights above- and below-ground, etc.) and the antioxidant enzyme system reflect that “FQ119” has stronger drought tolerance. Through RNA-seq analysis, a total of 68,316 transcripts (37,908 genes) were obtained. The largest number of significant differentially expressed genes (DEGs) in the comparison of “FQ118” and “FQ119” was observed at 6 h and 48 h. KEGG analysis demonstrated the significant enrichment of certain pathways associated with drought stress, such as glycerolipid metabolism and galactose metabolism. Co-expression analysis revealed that 7 hub DEGs, including genes encoding a photosystem reaction center subunit protein, chlorophyll a-b binding protein, glyceraldehyde-3-phosphate dehydrogenase A (GAPDH), and others, were coenriched in both comparisons. In addition, three hub genes specific to the comparison during the 6-h processing stage, encoding oxygen-evolving enhancer protein 1, receptor-like serine/threonine-protein kinase and calcium-transporting ATPase, were identified. The above hub genes were related to plant resistance to drought stress, and RT‒qPCR verified that the overall magnitudes of the differences in expression between the two lines gradually increased over time. Virus-induced gene silencing (VIGS) experiments have demonstrated that GAPDH plays a relevant role in the drought resistance pathway. In addition, the differences in expression of 7 DEGs encoding transcription factors, including Dofs, WRKYs, MYBs, and MYCs, also tended to increase with increasing duration of drought treatment, as determined via qPCR. In summary, this study identified several valuable genes related to plant drought resistance by screening genes with differential transcription under drought stress. This in-depth gene mining may provide valuable references and resources for future breeding for drought resistance in tomato.

## 1. Introduction

Drought is one of the main abiotic stresses that plants frequently encounter during their growth and development, and is very harmful to plant growth and yield [1]. Therefore, the cultivation of drought-resistant plant varieties is important for improving drought resistance in plants. The mechanisms of drought resistance in plants are complicated, so it is difficult to cultivate drought-resistant plant varieties via traditional breeding methods [2]. However, modern molecular biology technology can be used to explore and utilize drought resistance-related genes and effectively improve drought resistance in plants [3]. To date, research worldwide has focused mainly on introducing drought resistance genes through genetic engineering, and a variety of drought-resistant transgenic plants have been obtained [4].

Known drought resistance genes include both regulatory genes (such as transcription factors (TFs)) and functional genes (encoding core proteins). The main pathways involved include key enzymes for the synthesis of osmoregulatory substances, enzymes related to active oxygen scavenging, and functional proteins that protect cells from water stress [5]. Osmoregulatory substances can regulate and maintain the balance of osmotic pressure inside and outside cells, reduce water loss in the plant, and alleviate the damage caused by drought stress. These substances include amino acids and their derivatives (such as proline (Pro), betaine, etc.), low-molecular-weight sugars (such as trehalose, fructan, etc.), and polyols (such as mannitol, etc.) [6]. When plants encounter drought stress, excessive reactive oxygen species (ROS) accumulate, resulting in oxidative stress, causing membrane lipid peroxidation, protein and nucleic acid damage, and membrane damage [7]. Plants have developed antioxidant protection systems to remove reactive oxygen species and reduce oxidative damage in response to stress [8]. These systems include nonenzymatic antioxidant protection systems (ascorbic acid (AsA), reduced glutathione (GSH), etc.) and enzymatic antioxidant protection systems (superoxide dismutase (SOD), catalase (CAT), proxidase (POD), ascorbate peroxidase (APX), etc.) [9]. In addition, plants possess functional proteins that directly protect cells from water stress, including various macromolecular protective factors, membrane proteins and proteases, such as late embryogenesis abundant protein (LEA) [10], aquaporin (AQP) [11], and heat shock proteins (HSP) [12].

Tomato (*Solanum lycopersicum* L.) is rich in vitamin C, folate, lycopene, potassium and other nutrients and is one of the most widely cultivated vegetable crops in the world. In the early stages of drought or under mild drought conditions, tomato roots first perceive water stress in the soil, which leads to an increase in the root surface area and a decrease in the shallow layer of roots. With stress intensification in the late drought period, the normal growth of tomato roots is significantly inhibited, with decreases in the root surface area, root length and lateral root number [13]. During this period, in drought-resistant tomato plants, drought tolerance genes mediate stress signaling pathways to induce a variety of ions, such as Ca^2+^, to enter and exit guard cells, reduce the dynamic changes in osmotic substances in guard cells, and maintain the morphology of guard cells. This allows the physiological and biochemical characteristics of whole plants, such as the stomatal closing frequency, transpiration rate, net photosynthesis rate, stomatal conductance and intercellular CO_2_ concentration, to remain stable [14]. In addition, drought stress can trigger ROS accumulation, and the ROS homeostasis facilitated by drought resistance genes can reduce oxidative stress in cells, lipid peroxidation, membrane structure destruction, and protein and nucleic acid degeneration [15].

Drought stress slows the growth and development rate of tomato plants, affects dry matter accumulation, flowering and fruit setting, and can even cause the death of the plant, resulting in considerable economic losses. Therefore, the exploration of drought resistance genes for tomato breeding is critical for the sustainable development of the tomato-growing industry [16]. In this study, the growth and physiological indices of the sensitive tomato line “FQ118” and drought-tolerant tomato line “FQ119” were evaluated, and the DEGs between these cultivars under drought stress were further screened via RNA-seq and subsequently analyzed and identified to provide valuable gene resources and a theoretical basis for the subsequent breeding of drought-resistant tomato.

## 2. Results

### 2.1. Growth Indicator Determination of “FQ118” and “FQ119” Under Drought Stress

Two tomato lines, “FQ118” and “FQ119”, which are sensitive and tolerant to drought stress, respectively, were treated with PEG-6000 for 48 h, and the plant height, stem diameter and dry and fresh weights of the above- and below-ground parts were measured. “FQ118” presented an obvious wilting state, whereas “FQ119” maintained relatively robust and healthy growth after simulated drought treatment (Figure 1).

Due to the different growth characteristics of the lines (Figure 2), the growth of “FQ118” was greater than that of “FQ119” at the 6 true leaf stage but, because of its sensitivity to drought stress, the growth indicators of “FQ118” clearly decreased after PEG-6000 treatment. For instance, in the “FQ118” line, at the 18-h and 24-h stages after the drought treatment, the stem diameter and plant height indicators of the plants showed a significant downward trend, respectively. However, in “FQ119”, no significant effect on plant height and only a slight decrease in stem diameter were observed at 48 h after treatment. The above-ground dry weight of “FQ118” showed an overall downward trend, especially during the 18 to 24 h period, where the downward trend was very obvious. The above-ground fresh weight measurements showed significant decreases of 44.04% and 29.26% compared with 0 h at 6 h and 48 h after treatment, respectively. The fresh weight of the below-ground parts also decreased significantly, by 36.11% at 18 h and 54.17% at 48 h, which compared with 0 h, respectively, in “FQ118”. In “FQ119”, as the duration of drought treatment progresses, the above-mentioned indicators show a generally stable trend. The differences between adjacent time points are not significant, further demonstrating the drought tolerance of “FQ119”.

### 2.2. ROS Indicators Determination of “FQ118” and “FQ119” Under Drought Stress

ROS accumulation is often associated with plant stress. Peroxidase activity can regulate ROS levels via a side reaction, and the malondialdehyde (MDA) content reflects the peroxidation level of the cell membrane [17]. The activities of the ROS scavenging enzymes SOD, POD and CAT showed irregular and dynamic changes in the two lines; in “FQ119”, the activities of these three enzymes showed a stable upward or downward trend, especially after 18 h of drought treatment, whereas, in “FQ118”, the trend of change seems to be completely irregular. In conclusion, for the determination of antioxidant enzymes such as POD and CAT, in “FQ119” after drought treatment (18 to 48 h), the activities showed an upward trend. Meanwhile, in “FQ119” after 18 h of drought treatment, the down-regulation trend of MDA reflected a smaller extent of membrane lipid peroxidation loss, which in turn indicated its drought tolerance (Figure 3).

### 2.3. Transcriptome Analysis of “FQ118” and “FQ119” Under Drought Stress

To identify genes involved in drought stress with differential expression between tomato “FQ118” (A) and “FQ119” (F), 30 samples (encompassing 3 independent biological replicates) were obtained from the control (0 h), referred to as A1/F1, and drought stressed (6 h, 18 h, 24 h, and 48 h) tomato plants, referred to as A2-5/F2-5, for RNA-seq (SUB14898066). The number of raw reads ranged from 37.78 to 43.40 million per sample. After low-quality reads were removed, the number of clean reads ranged between 37.77 and 43.38 million. The Q20 and Q30 values of each sample were no less than 98.835% and 97.025%, respectively, and the GC content was between 42.522% and 44.46%, indicating that the sequencing data were reliable. On average, 88.77% of the reads mapped to unique locations in the reference genome, and 4.31% mapped to multiple locations (Appendix A). A total of 69,063 transcripts (37,908 genes) were obtained from all the samples, including 34,688 known transcripts (34,688 genes) and 34,375 novel transcripts (12,810 genes) (Appendix A).

### 2.4. Analysis of DEG Response to Soil Drought Stress

The expression levels of each gene in various samples were obtained via quantization of the read count data, and differential expression analysis was conducted using DESeq2. A logFoldChange ≥ 1 and adjusted *p*-value < 0.05 were used to define significant DEGs. In the intragroup comparisons of “FQ118”, a total of 7372 DEGs were identified in A1_vs._A2 (4382 up-regulated and 2990 down-regulated), and 9644 DEGs were identified in A1_vs._A5 (5102 up-regulated and 4542 down-regulated), demonstrating a period of drought stress with more DEGs in “FQ118”. In the intragroup comparison of “FQ119”, a total of 6340 DEGs were identified in F1_vs._F2 (3576 up-regulated and 2764 down-regulated), representing a period with the most DEGs in “FQ119” under drought stress. In the comparison between “FQ118” and “FQ119”, a total of 7930 DEGs were identified in A2_vs._F2 (3893 up-regulated and 4037 down-regulated), and 8925 DEGs were identified in A5_vs._F5 (4027 up-regulated and 4898 down-regulated), which may be critical periods for drought resistance genes to respond to stress (Figure 4).

A Venn diagram revealed 3515 and 522 total overlapping DEGs common to all 4 periods after drought treatment in “FQ118” and “FQ119”, respectively. A total of 548 DEGs were identified in the intergroup comparison (A_vs._F) at different treatment stages (Figure 5).

### 2.5. GO and KEGG Pathway Analysis of DEGs

The A2_vs._F2 and A5_vs._F5 comparisons mentioned above may reflect the differences in gene expression profiles at key stages of tomato drought resistance. The top 20 enriched GO terms in the A2 vs. F2 comparison revealed that the terms related to stress tolerance were significantly enriched in biological processes, such as response to salt stress (15 DEGs), chlorophyll biosynthetic process (14 DEGs) and photosynthesis (45 DEGs). In the cellular component category, the terms related to the cytoplasmic, membrane and photosynthetic systems, such as integral components of the membrane (1390 DEGs), cytoplasm (392 DEGs), plasma membrane (226 DEGs) and cytosol (155 DEGs), were significantly enriched and, in the molecular function category, the DEGs were enriched in terms such as ATP binding (515 DEGs), oxidoreductase activity (118 DEGs), calcium ion binding (88 DEGs), transmembrane transporter activity (86 DEGs) and chlorophyll binding (32 DEGs), etc. In the A5 vs. F5 comparison, response to hydrogen peroxide (16 DEGs) was the most widely represented biological process term. In the cellular component group, 1601, 415, 285 and 162 DEGs were associated with the same terms as those enriched in the A2_vs._F2 comparison. Moreover, in the molecular function group, in addition to ATP binding (639 DEGs) and chlorophyll binding (28 DEGs), key enriched terms included receptor serine/threonine kinase binding (14 DEGs), UDP–glycosyltransferase activity (72 DEGs), and protein serine/threonine kinase activity (173 DEGs) (Figure 6).

KEGG pathway analysis identified 147 and 150 terms as significantly enriched in both comparisons, respectively (Figure 7). All of these terms were classified into metabolism, genetic information processing, cellular processes, environmental information processing and organismal system classes. In the A2_vs._F2 comparison, the top 20 significantly enriched terms among the DEGs were involved in metabolic pathways (1289 DEGs). In addition, 38, 33, 32 and 65 DEGs were involved in glycerolipid metabolism, galactose metabolism, photosynthesis-antenna proteins and circadian rhythm–plant, respectively, which are closely related to plant drought tolerance [18,19]. In the A5_vs._F5 comparison, most DEGs were involved in metabolic pathways (1515 DEGs) and secondary metabolite biosynthesis (865 DEGs). The number of DEGs associated with the glycerolipid metabolism and circadian rhythm–plant pathways increased to 44 and 82, respectively. Additionally, the MAPK signaling pathway was found to be the most significantly enriched pathway, with a total of 290 DEGs identified.

### 2.6. Analysis of Drought Resistance-Related Gene Expression Patterns and Establishment of Co-Expression Network

Venn analysis demonstrated that a total of 548 DEGs between “FQ118” and “FQ119” were found in all 5 periods examined. These DEGs were divided into three groups according to their expression statistics and functional clustering (Figure 8), containing 316, 41 and 191 DEGs, respectively. The expression of the genes in Group I clearly tended to increase with longer duration of drought stress, whereas the expression of the enriched genes in Group III tended to decrease.

According to the results of the above top 20 KEGG pathway analyses, the DEGs related to plant drought tolerance in Groups I and III were identified (Appendix A). In the A2_vs._F2 comparison, a total of 14 DEGs encoding glutathione S-transferases were identified, of which 9 were up-regulated. A total of 8 DEGs encoded glycosyl transferases, of which 6 were up-regulated. In addition, 12 DEGs encoded phosphofructokinases, of which 10 were up-regulated. Additionally, 4 DEGs each encoded malic enzymes and glucose-6-phosphate dehydrogenases, which tended to be down-regulated. In the A5_vs._F5 comparison, most of the above genes did not differ significantly; however, among the significantly enriched pathways, such as metabolic pathways, mitogen-activated protein kinase (MAPK) signaling pathway–plant and circadian rhythm–plant, a total of 8 DEGs encoded protein phosphatase 2C proteins, and 4 DEGs encoded aldehyde dehydrogenases, all of which tended to be up-regulated. In addition, 19, 8, 6 DEGs encoding WRKY, Dof and b-Zip TFs, and 8 DEGs encoding bHLH and MYB TFs, respectively, were identified in this comparison.

For the A2_vs._F2 and A5_vs._F5 comparisons, a total of 221 and 224 DEGs (logFC > 1.5) were analyzed for correlations, and gene pairs with correlation coefficients greater than 0.9 were mapped with Cytoscape ver. 3.10.2. A total of 609 edges were generated, and two groups of co-expression networks were established in the A2_vs._F2 comparison. According to the number of edges and weight statistics, no effective hub genes were predicted in the Group II network. In contrast, a total of 13 hub genes were found in the Group I network, and these DEGs encoded proteins that tended to be involved in the processes of response to light stimulus and photosystem regulation, such as photosystem reaction center proteins and chlorophyll a-b binding proteins. In addition, a total of 7 DEGs from the two networks, encoding photosynthetic NDH subunit, NADPH, fatty acid beta-oxidation multifunctional protein, protein FAF-like and receptor-like protein Cf-9, etc., exhibited more than 10-fold differences in expression (Figure 9a).

In the A5_vs._F5 comparison, a total of 797 edges were obtained to complete the co-expression network. Similarly, these genes were divided into 2 groups, in which a total of 4 hub genes were screened from Group I (including receptor-like protein kinase NCRK and Ca^2+^ transporting ATPase, etc.) and 9 hub genes were screened from Group B. The Group II hub genes were also involved in the photosynthetic process; indeed, 7 DEGs were the same as the hub genes in the A2_vs._F2 comparison. A total of 7 DEGs showed a 10-fold difference in expression between the two sets of networks, among which 5 were the same as those identified in the A2_vs._F2 comparison. The other 2 DEGs encoded a chlorophyll a/b-binding protein and a BTB/POZ domain-containing protein (Figure 9b).

### 2.7. Expression Trends of Drought Resistance-Related Genes Identified via RT‒qPCR

In combination with the construction of the co-expression network, the expression levels of selected hub genes between “FQ118” (A) and “FQ119” (F) were identified at different stages of drought treatment. The results revealed that 7 hub genes were screened together in the A2_vs._F2 and A5_vs._F5 comparisons, among which 4 DEGs (*Solyc02g069450.3*, *Solyc06g066640.3*, *Solyc06g082950.5* and *Solyc06g084050.4*) encoding photosystem reaction center subunits (proteins) tended to be up-regulated in all periods within 48 h after drought treatment (Figure 10a). The relative expression level of *Solyc06g084050.4*, encoding a photosystem II reaction center W protein, was 68.6 times greater in the “FQ119” than in the “FQ118” and was almost twice as high as that of the other 3 genes at 48 h, and the up-regulated expression level was 52.88 times greater than that in the untreated materials (0 h).

Another 3 co-enriched hub genes (*Solyc10g005050.3*, *Solyc04g009030.3* and *Solyc09g014520.3*) encode the proteins CURVATURE THYLAKOID 1B, GAPDH and chlorophyll a-b binding protein CP29.1, which are involved in electron transport and ATP formation in photosynthesis [20,21]. The qPCR results revealed that the levels of *Solyc10g005050.3* and *Solyc09g014520.3* tended to first increase, then decrease, and then increase over the 48 h of treatment (Figure 10b). The differential expression ratio of the *Solyc04g009030.3* gene between “FQ118” and “FQ119” gradually increased over time. At 48 h after treatment, the difference in expression levels of these three genes were 7.31, 12.64 and 6.52 times greater than the corresponding differences at the 0 h stage.

The specific hub genes associated with plant resistance to drought tolerance from respective groups were verified via qPCR. The results demonstrated that the *Solyc02g065400.3*, *Solyc11g069590.2*, and *Solyc01g096190.3* genes, encoding oxygen-evolving enhancer protein 1 (OEE1), receptor-like serine/threonine-protein kinase NCRK, and calcium-transporting ATPase, respectively, were also highly expressed after treatment relative to their respective comparison groups. *Solyc11g069590.2* and *Solyc01g096190.3* also tended first to increase, then decrease, and then increase over 48 h of treatment (Figure 10c). At 18 h and 48 h, the differential expression level of *Solyc11g069590.2* in the two lines increased by 8.14- and 8.22-fold compared with that at 0 h, while *Solyc01g096190.3* expression exhibited 4.33- and 4.75-fold differences. The differential expression level of *Solyc02g065400.3* continuously increased; at 48 h, the ratio between the two lines increased to 17.22.

TFs play important regulatory roles in plant abiotic stress [22]. In this study, KEGG and co-expression network analyses identified a total of 7 TFs with highly differential expression trends in the two lines (Figure 11), including 2 Dofs, 2 WRKYs, 1 bHLH-MYC gene, 1 MYB8 gene and 1 TCP gene. qPCR analysis revealed that the levels of the Dof-encoding genes *Solyc02g067230.3* and *Solyc02g088070.3* tended to increase continuously. At 48 h of stress, the expression levels of the two genes in the comparison between “FQ118” and “FQ119” achieved increases of 97.29- and 43.95-fold, respectively. A WRKY protein encoded by *Solyc05g015850.5* tended to increase, then decrease, and then increase again after drought treatment. At 18 h and 48 h after reprocessing, the increases were 18.58- and 57.64-fold, respectively. The other WRKY encoded by *Solyc02g021680.3* exhibited an overall increasing trend, and the difference in expression reached 51.10-fold at 48 h. Another 3 TFs, bHLH (MYC), MYB8 and TCP, encoded by *Solyc10g009270.3*, *Solyc12g096200.2* and *Solyc06g065190.1*, also tended to be up-regulated after drought treatment, with the increases reaching 15.54-, 9.39- and 6.29-fold, respectively, at 48 h.

### 2.8. Function Validation of Genes Related to Drought Resistance by VIGS

After the preliminary screening of DEGs and the analysis of their expression patterns under drought stress, in this study, the *Solyc04g009030.3* (GAPDH) and *Solyc02g065400.3* (OEE1), closely related to the regulation of oxidative stress, were constructed in the *pTRV2* silencing vector. The leaves of tomato “FQ119” lines were treated with Agrobacterium-mediated inoculation. After the *pTRV2-PDS* plants exhibited bleaching, the expression levels of the two genes were determined. Plants were selected with silencing efficiency above 50% for drought treatment and phenotypic observation conducted after 48 h (Figure 12a). The results indicated that, when *GADPH* was silenced, the whole plants showed wilting symptoms, with curled leaf margins and weakened drought tolerance; on the other hand, when *OEE1* was silenced, although the plants also presented wilting symptoms, the difference was not significant. In summary, *GADPH* in this study might play a key role in the drought tolerance process of tomato plants (Figure 12b).

## 3. Discussion

Drought is a conventional abiotic stress that strongly influences plant growth and development. Therefore, for tomato, valuable drought resistance genes are sought to guarantee tomato breeding goals, such as high yield and improved fruit quality. In this study, two tomato lines with different drought sensitivities, “FQ118” (sensitive) and “FQ119” (tolerant), were subjected to drought treatment and then examined using RNA-seq. Samples were collected at 5 time points (A1–A5 and F1–F5), and a total of 30 samples were screened for differentially expressed genes. In total, 34,688 known transcripts (34,688 genes) and 34,375 novel transcripts (12,810 genes) were collected for analysis. Among the intergroup comparisons, groups A2_vs._F2 and A5_vs._F5 received particular attention in this study because of the Venn analysis results and high number of DEGs under drought treatment.

The KEGG analysis revealed three coenriched DEGs in the glycerolipid metabolism pathway, encoding diacylglycerol acyltransferase 2, aldose reductase (AR) and diacylglycerol kinase 1 (DGK), with notably elevated expression levels (logFC_(A2_vs._F2)_ = 10.51, 7.47 and 4.32; logFC_(A5_vs._F5)_ = 2.90, 4.88 and 1.07, respectively) (Appendix A). In previous studies, diacylglycerol acyltransferase was shown to be involved mainly in fat metabolism and lipid deposition in tissues [23]. Although it has been found to play a relevant role in the resistance of *Brassica napus* to external biological stress, there have been few studies on the function of diacylglycerol acyltransferase in plant drought resistance [24]; therefore, this topic needs to be studied further. AR is often involved in the abiotic stress response in plants. Early studies have shown that AR is involved in resistance to rice drought stress through responsive ABA signaling [25]. Recent studies have shown that heterologous expression of *ZmAR1* in *A. thaliana* may improve the stress tolerance of transgenic plants by reducing the content of sorbitol and regulating the expression levels of certain stress-related genes [26]. Eight DGK members were identified in the apple genome, and a comparative genomic analysis of apple materials with different drought stress responses revealed that most of the DGK members were significantly up-regulated in the drought-tolerant cultivars, suggesting that they may play a role in plant resilience to environmental challenges. Another study has shown that allogeneic transformation of the *MpDGK7* gene into *A. thaliana* conferred enhanced drought resistance in the overexpressing strains [27]. Additionally, in the galactose metabolism pathway, most of the genes encoding galactinol synthase (GolS) proteins were significantly enriched, and 3 DEGs with the greatest fold differences (logFC_(A2_vs._F2)_ = 7.07, 3.54 and 3.31) were highlighted (Appendix A). The metabolic pathways associated with GolS play important roles in plant carbon assimilation storage and transport, biological and abiotic stress responses and other life processes [28]. Studies have shown that GolS is closely involved in plant regulation of temperature stress, mainly because it can increase the contents of inositol galactoside and raffinose in plants [29]. In *Petunia hybrida*, the transcription of *PhGolS1-1* is activated by the C2H2 zinc finger protein-encoding gene *PhZFP1*, which promotes the biosynthesis of inositol galactoside and raffinose and ultimately improves the cold resistance of plants [30]. In long-term and in-depth research on *GOLS* in maize, Zhao et al. reported that *GOLS* participated in the response to external high temperature and salt stress [31]. Recent studies have shown that the overexpression of *ZmRAFS* can increase GOLS activity and that increasing the raffinose content can significantly maintain the physical water holding capacity of leaves, thereby improving drought resistance [32]. Moreover, GOLS can also participate in the resistance of plants to waterlogging through regulation of the sugar content [33].

Co-expression analysis revealed 12 and 13 core genes in the A2_vs._F2 and A5_vs._F5 comparison groups, respectively. Most of the genes encode proteins associated with photosynthesis signals. Seven DEGs were coenriched, including four DEGs encoding photosystem reaction center subunit proteins, one encoding the protein CURVATURE THYLAKOID 1B, one GAPDH, and one encoding the chlorophyll a-b binding protein CP29.1. Under salt, high-temperature and drought stresses, the photosystem reaction center subunits (proteins) can reduce cell water potential by accumulating proline. It can also regulate rapid transmembrane transport of cell water and maintain cell turgor pressure, indirectly ensuring the stable water environment required by the photosystem reaction center. Finally, it can reduce oxidative damage to the photosystem reaction center (PSI/PSII) by eliminating the accumulation of ROS triggered by stress, protect the stability of key structures, such as light-harvesting protein complexes (LHC) and Rubisco enzymes, and ultimately exert stress resistance and stress tolerance [34]. In previous studies, researchers used RNA-seq to screen DEGs in alfalfa under cold stress and reported that the genes encoding the photosystem I reaction center subunit and chlorophyll A-B binding protein in resistant lines were mostly up-regulated [35]. In peach, 19 genes encoding chlorophyll A-B binding proteins were identified in the genome database. RNA-seq analysis and RT‒qPCR revealed that the expression of nine genes was significantly up-regulated under drought stress, suggesting that these genes likely play important roles in the regulation of drought stress [36]. Another hub gene encodes the protein CURVATURE THYLAKOID 1B, which is involved in thylakoid membrane bending [37]; previous studies have shown that this protein plays a role in controlling photosynthetic electron transport [38]. In this study, the difference in the expression of the above hub genes between the “FQ118” and “FQ119” lines tended to increase over time under drought stress, and these genes may promote the stability of the photosynthetic system to ensure material synthesis and maintain normal plant metabolism [39]. One hub gene (*Solyc04g009030.3*) encoded GAPDH, which is a key mediator in many oxidative stress responses and is involved in nuclear translocation and cell death induction [40]. Under biological stress, GAPDH can promote the binding of coat protein to ATG3, suggesting that this complex may function to inhibit autophagy [41]. In the context of abiotic stress, studies have shown that GAPDH accumulates in the nucleus under heat stress, and *GAPDH* overexpression increased heat tolerance in Arabidopsis seedlings [42]. In this study, the difference in expression of the GAPDH encoding gene *Solyc04g009030.3* between “FQ118” and “FQ119” gradually increased. At 48 h of drought stress, the difference was 12.64 times greater than that at the 0 h stage. In previous research on drought stress, 13 GAPDH-encoding genes in *Quercus rubra* were identified through genomic biogenic screening. The determination of the expression levels of target proteins and genes in different tissues under drought stress suggests that parts of *QrGAPDH* may play a key role in the response of trees to drought stress [43]. In another study, four *GAPDH* genes were screened from drought-resistant line of *Gossypium hirsutum* via RNA-seq, and silencing *GhGAPDH9* was observed to cause significant leaf wilting or whole-plant dieback after drought stress [44]. Therefore, *Solyc04g009030.3*, identified via VIGS in the present study, is likely involved in the tomato response to drought stress and may have high application value for improving tomato drought resistance.

Three hub genes from each comparison encode oxygen-evolving enhancer protein 1, receptor-like serine/threonine-protein kinase NCRK and calcium-transporting ATPase, respectively. There were differences in the expression of these genes between “FQ118” and “FQ119” at each stage after drought treatment, and the magnitude of the difference tended to increase over time. The oxygen-evolving enhancer protein is a major factor in reactive oxygen species pathways and is also widely involved in photosynthetic signaling in plants [45,46]. Previous studies in grapevine have shown that this protein is the target of the conserved effector secretory protein RXLR31154 in *P. viticola* and that RXLR31154 affects the photosynthetic system by mediating the activity of this protein, thereby affecting grape resistance to *P. viticola* [47]. Similarly, oxygen-evolving enhancer proteins affect the resistance level of mandarins to *Candidatus* Liberibacter asiaticus by participating in the regulation of photosynthetic signals [48]. In terms of crop drought and salt tolerance, RNA-seq analyses of different crops have revealed the differential expression of genes encoding oxygen-evolving enhancer proteins and key genes affecting photosynthetic signals, but the specific mechanism involved in the above abiotic stress pathways remains unknown [49,50,51]. Calcium-transporting ATPase plays a key role in important signal transduction processes in plant cells [52]. NCRK influences the transcription of downstream genes, mainly by receiving extracellular signals, and participates in many biological pathways, including resistance and disease resistance [53]. Therefore, the hub genes identified in this study, encoding the important proteins above, are likely to participate in the regulation of drought stress in plants, and their specific functions need to be further verified.

TFs often play important regulatory roles in crop growth, development and resistance to stress and disease [54]. Protein‒protein interaction networks and RT‒qPCR analyses in tea have revealed that most *CsDof* genes respond to drought stress [55]. In potato, *StCDF1* and a long-chain noncoding RNA (lncRNA) called *StFLORE* also regulate water loss by affecting stomatal growth and diurnal opening [56]. MYBs, WRKYs and MYCs are large TF families in plants that often play vital roles in regulating drought resistance, salt tolerance, low-temperature tolerance, etc., in crops [57,58,59]. TCPs are often involved in the formation of developmental organs in crop plants [60]. The expression of *ZmTCP42* in maize is reportedly induced by ABA, and the overexpression of this gene in *A. thaliana* can lead to hypersensitivity to ABA during seed germination and improve drought resistance [61]. In this study, 7 DEGs with significantly differential expressions, encoding the above TFs were screened via RNA-seq, and all of them presented an overall increasing trend in expression. Notably, the differences in the expression of 2 *Dofs* and 2 *WRKYs* were greater than 40-fold at 48 h of drought treatment according to qPCR, indicating that these genes may play a regulatory role in tomato drought resistance. Therefore, in-depth functional mining of these TFs will be highly important for improving the drought resistance of tomato germplasm.

## 4. Materials and Methods

### 4.1. Plant Materials and Growth

Seeds of the drought-tolerant tomato “FQ119” and drought-sensitive “FQ118”, with full grains and of the same size, were selected. The seeds were washed with distilled water 2–3 times, soaked in warm soapy water at 55 °C for 15 min, and then planted in a pot. The seedling substrate ratio was 3:1:1 soil/perlite/vermiculite. The culture conditions included an ambient temperature of 25 °C, a relative air humidity of 60%, and a 16 h/8 h diurnal photoperiod. At the common stage of growth and development, the plant of “FQ119” is smaller in size compared to that of “FQ118”.

### 4.2. Drought Treatments and RNA-Seq Assay

When the tomato plants to be tested reached the stage with six true leaves, 20% PEG-6000 solution was dripped slowly along the inner wall of the flowerpot in a circular direction using a drip pipe. Meanwhile, the control group was treated with the same amount of distilled water. After the treatment was completed and the plants were left to stand for 10 min, the tested plants were returned to the culture environment for subsequent index measurements. Tomato leaves were collected at 0 h (CK), 6 h, 18 h, 24 h and 48 h after stress treatment, in three biological replicates per stage, followed by RNA-seq (Illumina 2500 (Illumina, San Diego, CA, USA); HiSAT2 used to map the reads to the genome, and HTSeq and DESeq used for gene expression statistics and DEG analysis) and other experiments.

### 4.3. Construction of RNAi Vectors

The CDS of target genes was obtained from the SGN database (https://solgenomics.net (accessed on 11 March 2024)), total RNA was extracted from ‘Moneymaker’ tomato leaves using a reagent kit (Vazyme, Nanjing, China), and reverse-transcribed cDNA was obtained using a reagent kit (Vazyme, Nanjing, China). Using *Bam*H I as the cleavage site, specific upstream and downstream primers with a 20 bp recombinant homologous arm were designed, and the high-fidelity enzyme Phanta Max (Vazyme, Nanjing, China) was used for PCR amplification of the CDS region.

### 4.4. RNAi Line Construction

The recombinant *pTRV2* target genes, unloaded *pTRV2*, and *pTRV2-PDS* vectors were subsequently transformed into Agrobacterium GV3101, which was injected into the leaves of “FQ119” tomato lines at the four-leaf and one-heart stage, with 30 plants treated with each. After infection, the plants were first stored at 18–21 °C. The tomatoes were cultured in darkness with a humidity greater than 80% for 72 h and then cultured under a photoperiod of 16 h/8 h at 23~25 °C and 60% humidity. After the *pTRV2-PDS* plants exhibited bleaching (the *PDS* gene is involved in the biosynthesis of carotenoids in plants and silencing of this gene leads to the absence of photoprotective pigments in chloroplasts, causing photodamage and eventually resulting in albino or bleached phenotypes of leaves after approximately 2 weeks), the expression level of target genes was measured, and tomato plants with a silencing efficiency greater than 50% were selected for follow-up tests.

### 4.5. RT‒qPCR

Total RNA was extracted from tomato leaves via an OminiPlant RNA Kit (CW2598S, Cowin Biotech, Taizhou, China), and cDNA was obtained via reverse transcription (R211-01 kit, Vazyme, Nanjing, China). The RT‒qPCR reactions were prepared as follows: cDNA, 2 µL; Vazyme ChamQ SYBR qPCR Master Mix, 10 μL; primer (F), 0.4 μL; primer (R), 0.4 μL; ddH_2_O, 7.2 μL. The PCR program was as follows: pre-denaturation at 95 °C for 3 min; denaturation at 95 °C for 30 s and annealing at 58 °C for 30 s for a total of 40 cycles; and extension at 72 °C for 30 s. Three technical repeats were performed for each reaction, three wells with Ct value differences less than 0.5 were selected as test data, and the relative gene expression was calculated by the 2^−ΔΔCT^ method [62].

### 4.6. Growth Indicator Determination

Tomato plants at the sixth true leaf stage were selected and treated with 20% PEG-6000. Plant height and stem diameter were measured at the 0 h (CK)-48 h stage. At the same time, the plants were deoxidized at 105 °C for 15 min, and the dry and fresh weights of the above-ground and below-ground parts were measured during the treatment stage.

### 4.7. Physiological Indicator Determination

Tomato plants at the sixth true leaf stage were selected and treated with 20% PEG-6000. During the 0–48 h drought stress stage, the SPAD value was monitored via a chlorophyll analyzer (SPAD-502, Konica Minolta, Tokyo, Japan); the enzyme activities of SOD, POD, and CA were determined via a kit (G0101W, G1212W, G0106W, Geruisi, Beijing, China); and the Pro and MDA contents were determined via a kit (ml093018 and ml094964, Mlbio, Shanghai, China). The soluble sugar content was determined according to the Ma method [63].

## Figures and Tables

**Figure 1 plants-14-01471-f001:**
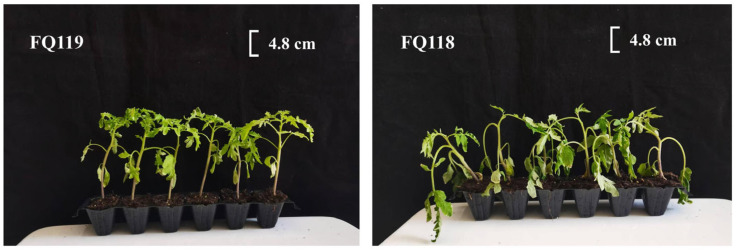
Growth state of “FQ118” and “FQ119” after PEG-6000 treatment for 48 h.

**Figure 2 plants-14-01471-f002:**
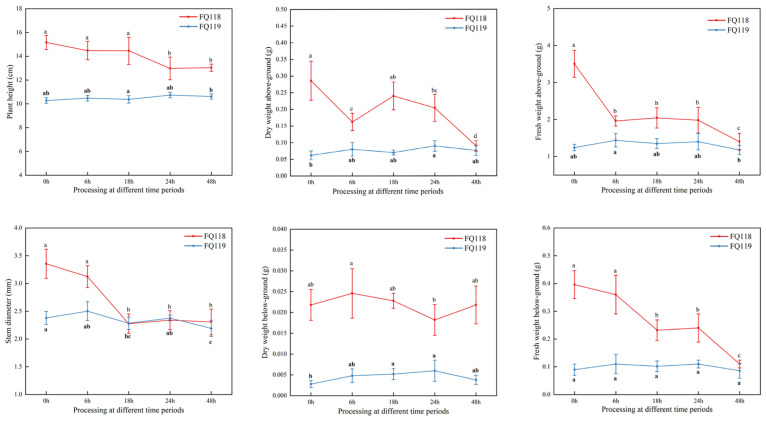
Growth indicator determination of “FQ118” and “FQ119” after PEG-6000 treatment. The results are presented as the means ± SEM (*n* = 3, *p* < 0.05), through *t*-test statistics. The significant differences among time periods within a line are marked by a, b, c and d, respectively.

**Figure 3 plants-14-01471-f003:**
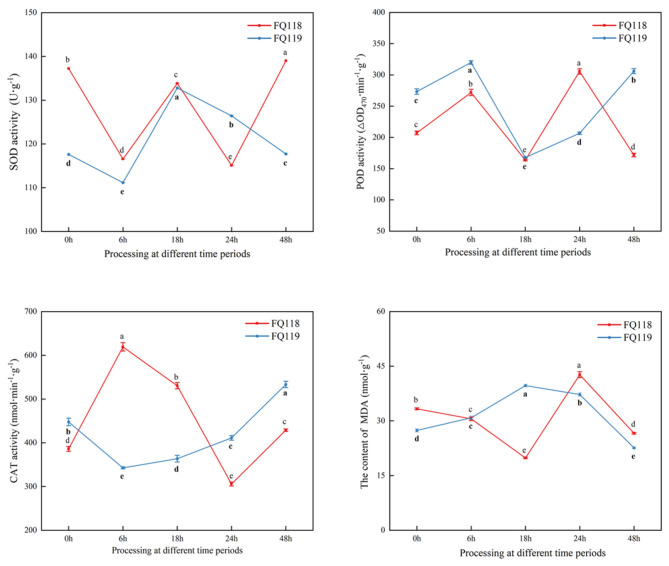
Antioxidant enzyme activities and MDA content determination of “FQ118” and “FQ119” after PEG−6000 treatment. The results are presented as the means ± SEM (*n* = 3, *p* < 0.05), through *t*-test statistics. The significant differences among time periods within a line are marked by a, b, c, d and e, respectively.

**Figure 4 plants-14-01471-f004:**
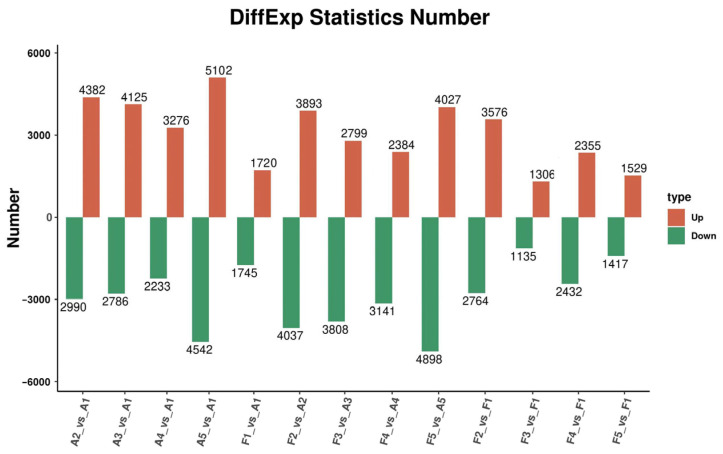
DEGs statistics between different comparison groups in RNA-seq of “FQ118” and “FQ119” after drought treatment.

**Figure 5 plants-14-01471-f005:**
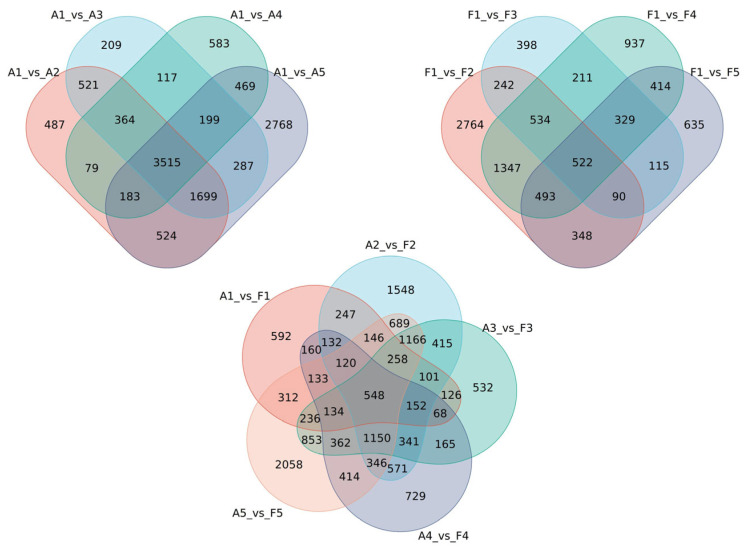
Venn statistics from RNA-seq of “FQ118” and “FQ119” after drought treatment.

**Figure 6 plants-14-01471-f006:**
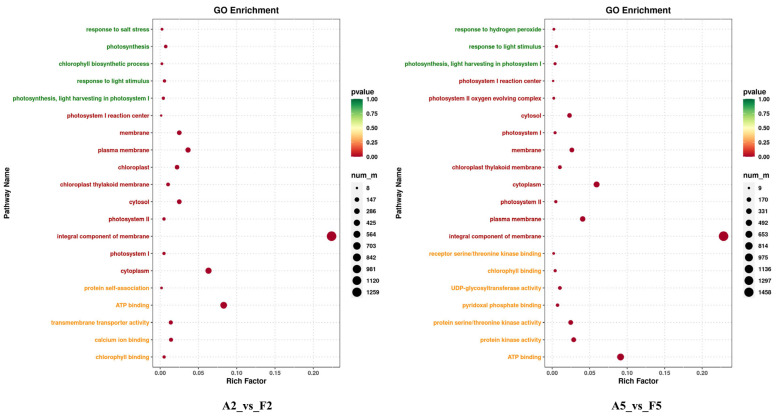
Top 20 GO terms of comparisons of A2_vs._F2 and A5_vs._F5. The green, red, and yellow descriptions in ordinate represent categories of biological processes, cellular component, and molecular function, respectively. The abscissa represents the level of the enrichment factor. The size of the circles in different terms in the analysis represents the number of DEGs, and the depth of the color represents the level of significance.

**Figure 7 plants-14-01471-f007:**
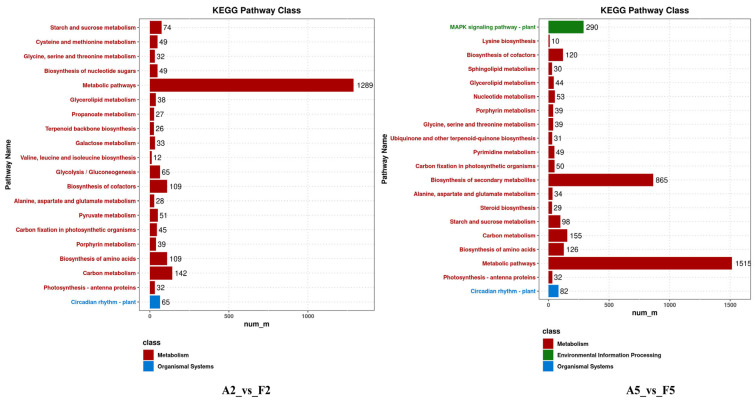
Top 20 KEGG terms of comparisons of A2_vs._F2 and A5_vs._F5.

**Figure 8 plants-14-01471-f008:**
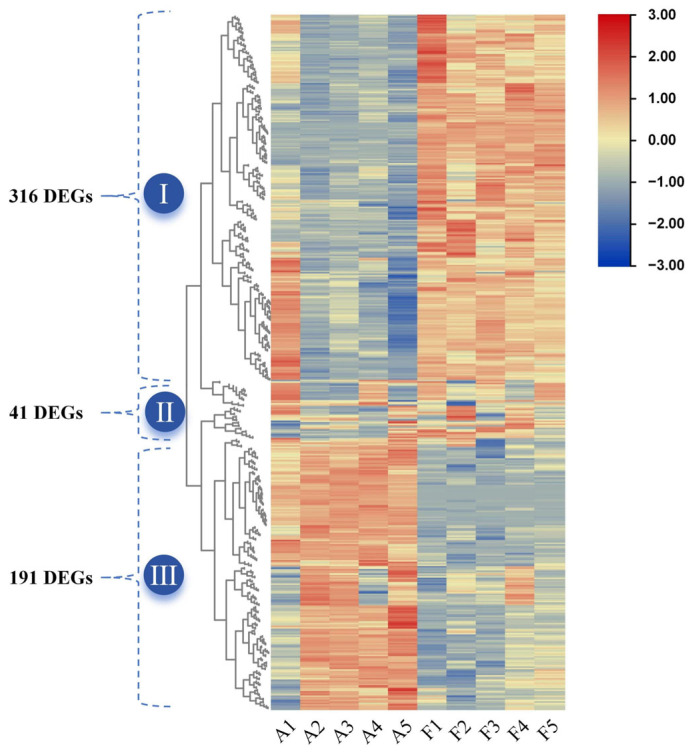
Analysis of the expression trend of common DEGs of “FQ118” and “FQ119” in 5 stages after drought treatment. The red and blue colors of the module genes, respectively, represent the trends of up-regulation and down-regulation of expression.

**Figure 9 plants-14-01471-f009:**
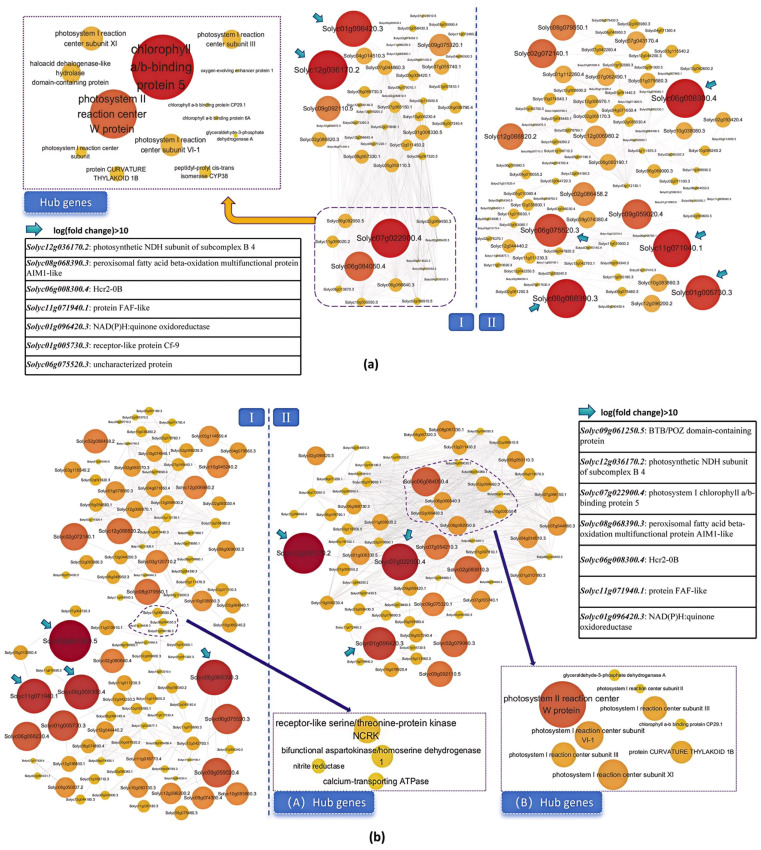
The co-expression analysis network of comparison groups A2_vs._F2 and A5_vs._F5. The circles represent genes and, the darker the color, the larger the trait, and the higher the multiple of difference. (**a**) Co-expression analysis of A2_vs._F2. According to the edge connection, this is divided into two subgroups I and II. According to the number of correlations (>0.9) and edges (>30), the hub genes (inside the purple dotted line) are screened out, with arrows indicating DEGs with a difference logFC>10 or more. (**b**) Co-expression analysis of A5_vs._F5.

**Figure 10 plants-14-01471-f010:**
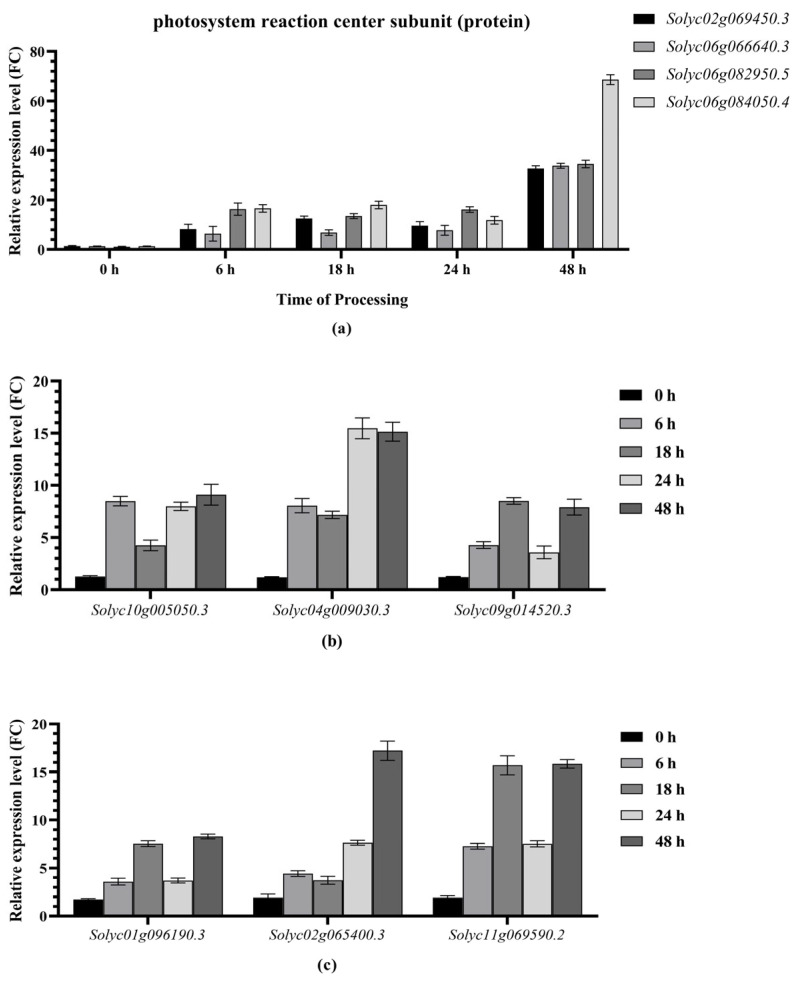
Comparative analysis of differential expression of hub genes between “FQ118” and “FQ119” in co-expression network via RT-qPCR. The X-axis of (**a**) represents the inoculation time of different genes, while those of (**b**,**c**) represent different genes. The Y-axis of all figures represents the relative expression level, with the unit being fold change (FC).

**Figure 11 plants-14-01471-f011:**
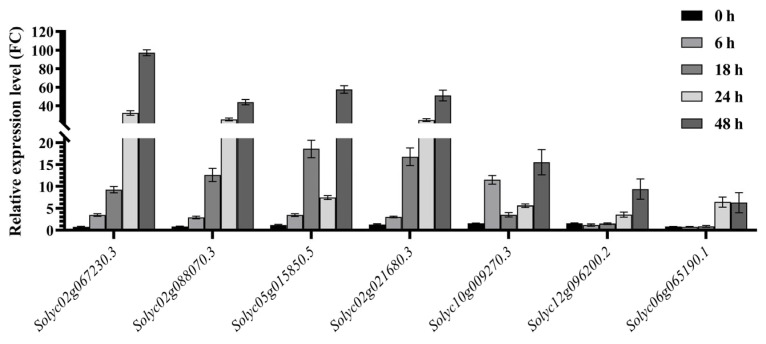
Comparative analysis of differential expression of DEGs encoding TFs between “FQ118” and “FQ119” in co-expression network via RT-qPCR. The X-axis represents different genes and the Y-axis represents the relative expression level, with the unit being fold change (FC).

**Figure 12 plants-14-01471-f012:**
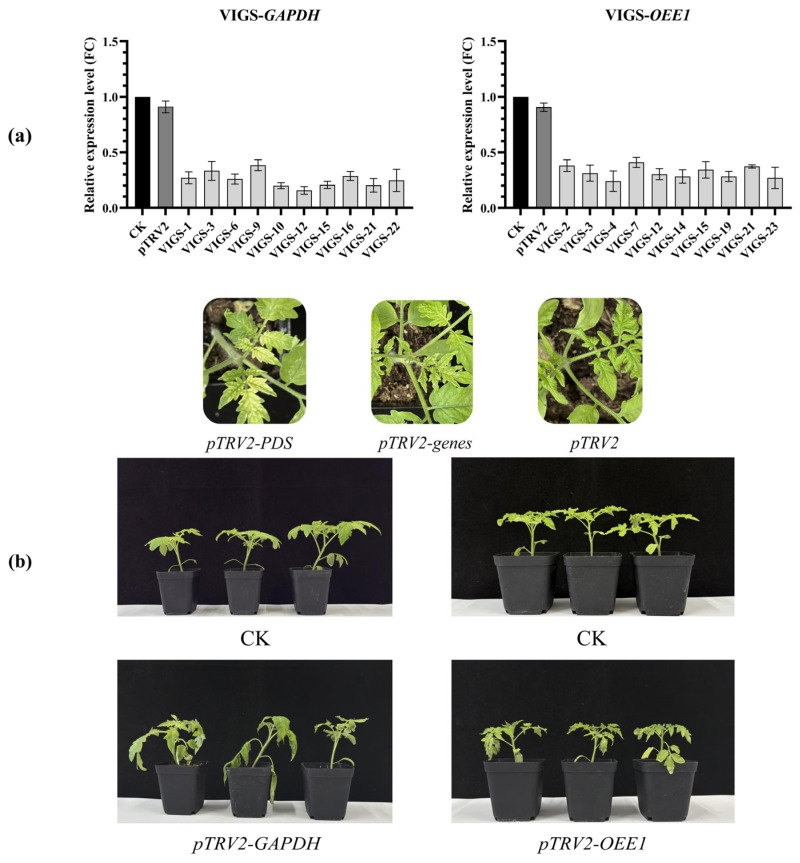
Identification of drought treatment phenotypes after silencing *GADPH* and *OEE1* genes in tolerant line “FQ119”. (**a**) represents the assessment of the silencing efficiency of two genes, the X-axis represents different silent plants and the Y-axis represents the relative expression level, with the unit being fold change (FC). (**b**) indicates the phenotypic identification of the silent lines during the 48-h stage of drought treatment.

## Data Availability

The raw data supporting the conclusions of this article will be made available by the authors on request.

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
