# Peer review of "Screening and Identification of Drought-Tolerant Genes in Tomato (Solanum lycopersicum L.) Based on RNA-Seq Analysis"

_plants, 2025, doi:10.3390/plants14101471_

Round 1
Reviewer 1 Report
Comments and Suggestions for Authors
The authors compare the gene expression changes of drought sensitive and drought resistant tomato varieties using PEG-6000 to mimic water stress. Plant appearance and other biological measurements indicated the FQ119 cultivar was more resistant to drought. Comparison of the sensitive FQ118 and resistant FQ119 cultivars revealed numerous gene expression changes that could be important for tomato drought tolerance. The included transcription factors and gene involved in glycerol biosynthesis.
A few suggestions below to improve the manuscript,
The experiments should be done with FQ119 and FQ118 plants grown in optimal water to have a real comparison of changes in gene activity due to the PEG-6000 induced water stress and have appropriate controls
The data in Figure 2 would suggest that the FQ119 cultivar produces smaller plants, and this should be noted in the introduction
There are no error bars on the Figure 3 antioxidant enzyme activities’ graphs and it is difficult to identify true trends within this data, the authors should pull back and temper their conclusions based on this data
Line 156, would it be more useful to apply a logFoldChange ≥ 2 and adjusted p-value < 0.05 to define significant DEGs?
Lines 212-213 and Figure 7, Were any of the upregulated genes included in the high osmolarity glycerol (HOG) pathway and the phosphatases that regulate this pathway (e.g. those genes/protein described in S. cerevisiae)? If so, they could be specifically mentioned in the discussion due to differences in the timing of the changes in gene expression the authors observed
The authors should try to distinguish among cause and effect gene expression changes, e.g. Figure 10 “photosystem reaction center subunit changes are likely a result of the osmotic stress rather than the plants attempt to adjust to the water stress, while Figure 11 gene expression changes may be more relevant to the findings
Did the authors perform the agrobacterium silencing experiments in the FQ119 cultivar as well? This may help confirm the findings of this project.
Author Response
Dear Editor and Reviewer 1:
Thank you for your letter and for the reviewer’s comments concerning our manuscript entitled “Screening and Identification on Drought-Tolerant Genes in Tomato (Solanum lycopersicum L.) Based on RNA-seq Analysis”. (ID: plants-3596571). Those comments are all valuable and very helpful for revising and improving our paper, as well as the important guiding significance to our researches. We have studied comments carefully and have made correction which we hope meet with approval. Revised portion are marked in the paper. The main corrections in the paper and the responds to the reviewer's comments are as flowing:
A few suggestions below to improve the manuscript:
1. The experiments should be done with FQ119 and FQ118 plants grown in optimal water to have a real comparison of changes in gene activity due to the PEG-6000 induced water stress and have appropriate controls
Response 1: Many thanks to the reviewer’s suggestions on the manuscript. In this study, two tomato materials with different sensitivity to drought stress, namely 'FQ118' and 'FQ119', were selected as the experimental materials. A control (0 h) without PEG treatment and under normal water supply conditions was set up. This point was mentioned in the method descriptions of lines 511-512 and 548-549.
2. The data in Figure 2 would suggest that the FQ119 cultivar produces smaller plants, and this should be noted in the introduction
Response 2: Considering the reviewer's suggestion, we have added descriptions about the growth state of two of tomato materials (line 96-97).
3. There are no error bars on the Figure 3 antioxidant enzyme activities’s graphs and it is difficult to identify true trends within this data, the authors should pull back and temper their conclusions based on this data
Response 3: Thanks to the reviewer's valuable suggestions. The determination of each indicator in Figure 3 of this study actually has error bars, but the differences in the values between repetitions are not significant, and the circle shape of the values is too large, making the error bars hardly visible. We are also grateful for the reminder from reviewer. Therefore, we have reorganized and redrawn this figure (line 152), and provided the relevant raw data in the attachment. We hope this can improve the efficiency of interpreting our results in this part.
4. Line 156, would it be more useful to apply a logFoldChange ≥ 2 and adjusted p-value < 0.05 to define significant DEGs?
Response 4: Thanks very much for the reviewer's query. Because the standard for filtering DEGs on the sequencing platform is based on logFoldChange≥1, and also to identify some genes with modest up-regulation but still play crucial roles. However, in actual analysis, whether it is for clustering expression patterns or WGCNA identification, we always conduct the screening from high to low based on the expression differences. Therefore, our experimental data are reliable and accurate.
5. Lines 212-213 and Figure 7, Were any of the upregulated genes included in the high osmolarity glycerol (HOG) pathway and the phosphatases that regulate this pathway (e.g. those genes/protein described in S. cerevisiae)? If so, they could be specifically mentioned in the discussion due to differences in the timing of the changes in gene expression the authors observed
Response 5: Thanks to the reviewer's valuable suggestions. We conducted significant enrichment analysis of the selected DEGs in KEGG pathways. It was found that none of the pathways related to the regulation of HOG and phosphatases mentioned by the reviewer were enriched.
6. The authors should try to distinguish among cause and effect gene expression changes, e.g. Figure 10 “photosystem reaction center subunit changes are likely a result of the osmotic stress rather than the plants attempt to adjust to the water stress, while Figure 11 gene expression changes may be more relevant to the findings
Response 6: Thanks to the reviewer's valuable suggestions. According to the reviewer's opinions, we have strengthened the description of photosystem reaction center subunit (line 414-424) in the discussion section from the perspective of osmotic adjustment. The genes encoding photosystem reaction center subunit that showed these differences were identified through WGCNA. Therefore, we selected two core genes for VIGS validation in the later stage. Indeed, as the reviewer pointed out, the transcription factors in Figure 11 play an important regulatory role in crop drought resistance. In the later studies, the regulatory targets of these transcription factors will be screened and identified, and their molecular mechanisms will be analyzed.
7. Did the authors perform the agrobacterium silencing experiments in the FQ119 cultivar as well? This may help confirm the findings of this project.
Response 7: Thanks very much for reviewer's query. The VIGS strategy adopted in this study was carried out in the drought-resistant material 'FQ119', which was mentioned at line 345-349. By silencing the key core genes that need to be detected in the drought-resistant material, differences in phenotypes were observed, which laid a solid foundation for the in-depth exploration of the functions of the two genes in the future.
Thanks again to the reviewer for taking time out of their busy schedules to review our manuscript. As suggested, we have developed the quality of English language throughout the whole manuscript. We hope that the above changes can improve the level of our manuscript to meet the standards of journal publication.
Reviewer 2 Report
Comments and Suggestions for Authors
The genomics portion of this manuscript seems to have been well done. However, the portion describing plant response to PEG (drought simulation) is poorly analyzed, described, and presented, making review very difficult. Some conclusions do not seem to be supported by the data, but it is possible that this is due to a very poor and unclear description and presentation. More details are below:
Lines 15-16: If FQ118 and FQ119 are cultivars, then they would have single quotation marks (not double). But I’m not sure that they are cultivars, especially since the authors say that they are “material.” Also try to be more clear than just “material.” Is it appropriate to use the word “line”?
Lines 22-23: Try to avoid undefined abbreviations as much as possible. Most often you will want to define the abbreviation at first use in the abstract and main body. Here, I recommend “…significant differentially expressed genes (DEGs)…” Also, here in the abstract, “A2_vs_F2” means nothing to the reader at this point. You will need to use more decriptive wording. There are many other undefined abbreviations used throughout the manuscript, which I am not specifically noting here in the review, but should be defined at first use (such as “VIGS” and “MDA”). If the abbreviation is very common, such as “RNA” or “RT-qPCR” then of course defining the abbreviation is not necessary.
Lines 101-102: Please provide a citation for the difference in drought tolerance for FQ118 vs. FQ119.
Lines 111-112: The authors state that FQ118 had significant reduction in height at 18 h, but in Fig. 2, FQ118 data points for height are labeled “a” for both 0 h and 18 h. Doesn’t this mean that there isn’t significant decrease in height at 18 h?
Lines 114-115: Is the 43.46% decrease in aboveground dry mass for FQ118 at 6 h an outlier (looking at Fig. 2)? Or is there a reason why dry weight went up significantly from 6 h to 18 h. If an outlier, it’s probably best to not focus on the 43.46% as relevant.
Line 117: I don’t think the 29.26% decrease in aboveground fresh weight for FQ118 at 48 h is correct, according to Fig. 2, because the aboveground fresh weight at 48 h is even less than the weight at 6 h, which was a 44.04% reduction. I’m assuming that all percentage calculation are based on the 0 h data?
Figure 2: Please indicate what the letters for each point are. I assume they designate significance, but only across times for each type of tomato, and not between types of tomato? This needs to be made clear. Also, “down-ground” should be “below-ground” in two of the figures (or “belowground” to be consistent with the Results text).
Line 118: Looking at Fig. 2, belowground fresh weight at 48 h is more than the 54.17% stated by the authors. Again, I am comparing 48 h to 0 h. But it looks like the authors may be comparing 48 h to 24 h? Please double-check calculations and when stating % reduction, please calculate the reduction based on the control/check at 0 h.
Line 119: What is meant by “any of these factors”. Does it only refer to what was noted about FQ118 in the text, or does it mean all the data given in Figure 2? Also, why only look for significance at “adjacent time points”? This is misleading because it suggests that there is no significance across the experiment for FQ119. For example, stem diameter of FQ118 at 18 h is significantly less than at 0 h (using the letters, which I assume indicate significant differences). In fact stem diameter for FQ119 at 48 h is significantly less than at 24 h, which would be “adjacent time points.”
Lines 128-130: I agree that after 18 h, the trends for FQ119 seem somewhat “stable.” But the authors state that an “opposite trend was observed” for FQ118. However, looking at Figure 3, the data points for FQ118 look erratic and I don’t think the authors should be confident about any trends for FQ118, even after 18 h. For POD, isn’t there an increase from 18 h to 48 h (including 24 h), for FQ118, like (but not as clear as) FQ119?
Figure 3: The caption for Figure 3 indicates that standard error is shown, but it is not shown in the figure. I think the standard error bars should be added.
Lines 228-240: When the authors refer to genes that are “up-regulated” or “down-regulated” it is not always clear if they mean up/down in FQ118 or FQ119. Please clarify.
Line 330 (and/or line 490): I assume that for pTRV2-PDS, the “PDS” means RNA interference for phytoene desaturase, which would be expressed as bleaching, and this is used as a marker? Please clarify for readers who aren’t familiar.
Line 471: Were seeds sprouted in a “dish” and then transplanted into a pot? Or should the word “dish” be changed to “pot”?
Lines 476-477: How was PEG “added to the roots”?
Comments on the Quality of English LanguageThe English is fairly good, but still needs some improvement for clarity. Also, many abbreviations need to be defined.
Author Response
Dear Editor and Reviewer 2:
Thank you for your letter and for the reviewer’s comments concerning our manuscript entitled “Screening and Identification on Drought-Tolerant Genes in Tomato (Solanum lycopersicum L.) Based on RNA-seq Analysis”. (ID: plants-3596571). Those comments are all valuable and very helpful for revising and improving our paper, as well as the important guiding significance to our researches. We have studied comments carefully and have made correction which we hope meet with approval. Revised portion are marked in the paper. The main corrections in the paper and the responds to the reviewer's comments are as flowing:
The genomics portion of this manuscript seems to have been well done. However, the portion describing plant response to PEG (drought simulation) is poorly analyzed, described, and presented, making review very difficult. Some conclusions do not seem to be supported by the data, but it is possible that this is due to a very poor and unclear description and presentation. More details are below:
1. Lines 15-16: If FQ118 and FQ119 are cultivars, then they would have single quotation marks (not double). But I’m not sure that they are cultivars, especially since the authors say that they are “material.” Also try to be more clear than just “material.” Is it appropriate to use the word “line”?
Response 1: Many thanks to the reviewer’s suggestions on the manuscript. Indeed, FQ118 and FQ119 are not cultivars. According to the requirements, we have replaced the incorrect description such as "material" with "line" and made relevant revisions throughout the text (As for the line 16 and line 33, etc.).
2. Lines 22-23: Try to avoid undefined abbreviations as much as possible. Most often you will want to define the abbreviation at first use in the abstract and main body. Here, I recommend “…significant differentially expressed genes (DEGs)…” Also, here in the abstract, “A2_vs_F2” means nothing to the reader at this point. You will need to use more decriptive wording. There are many other undefined abbreviations used throughout the manuscript, which I am not specifically noting here in the review, but should be defined at first use (such as “VIGS” and “MDA”). If the abbreviation is very common, such as “RNA” or “RT-qPCR” then of course defining the abbreviation is not necessary.
Response 2: Considering the reviewer's suggestion, we have supplemented the abbreviations in the abstract section (line 22 and 34). Moreover, we have corrected the unclear descriptions proposed by the reviewer (line 158-162). On this basis, we have reviewed the full text and supplemented the undefined abbreviations (As for the line 254 and line 394, etc.).
3. Lines 101-102: Please provide a citation for the difference in drought tolerance for FQ118 vs. FQ119.
Response 3: Thanks to the reviewer's valuable suggestions. In fact, this is the first time that we have conducted a groupomics analysis and phenotypic identification on these two lines FQ118 and FQ119 . We have not published any related papers. Therefore, we have no literature to support our work, which is why we conducted the 48-hour drought treatment observation (Figure 1).
4. Lines 111-112: The authors state that FQ118 had significant reduction in height at 18 h, but in Fig. 2, FQ118 data points for height are labeled “a” for both 0 h and 18 h. Doesn’t this mean that there isn’t significant decrease in height at 18 h?
Response 4: Thank you for the reviewer's comments. We are also very sorry that there was a problem with our description. In fact, what we want to convey is that the plant height and stem diameter have shown significant decreases respectively during the 18-hour and 24-hour drought treatment periods. In response to this, we have made some revisions in an effort to enhance readability (line 113-115).
5. Lines 114-115: Is the 43.46% decrease in aboveground dry mass for FQ118 at 6 h an outlier (looking at Fig. 2)? Or is there a reason why dry weight went up significantly from 6 h to 18 h. If an outlier, it’s probably best to not focus on the 43.46% as relevant.
Response 5: Thanks very much for reviewer's query. As pointed out by the reviewer, a sudden drop of 6 h might be an outlier, so we have changed the description of this sentence (line 118-120).
6. Line 117: I don’t think the 29.26% decrease in aboveground fresh weight for FQ118 at 48 h is correct, according to Fig. 2, because the aboveground fresh weight at 48 h is even less than the weight at 6 h, which was a 44.04% reduction. I’m assuming that all percentage calculation are based on the 0 h data?
Response 6: Thanks very much for reviewer's query. Indeed, the fresh weight indicators of the above-ground parts decreased at a rate compared with CK (0-hour) in the 6-hour and 48-hour drought treatments. The possible ambiguity in the description has been resolved by modifying this sentence (line 121-123).
7. Figure 2: Please indicate what the letters for each point are. I assume they designate significance, but only across times for each type of tomato, and not between types of tomato? This needs to be made clear. Also, down-ground should be “below-ground” in two of the figures (or “belowground” to be consistent with the Results text).
Response 7: Thanks to the reviewer's valuable suggestions. Indeed, different letters represent significant differences among the various drought treatment stages within each group. We have added this explanation in the figure caption (line 133-134 and line 155-156). Meanwhile, we have unified the descriptions of the “down-ground” and the “below-ground”.
8. Line 118: Looking at Fig. 2, belowground fresh weight at 48 h is more than the 54.17% stated by the authors. Again, I am comparing 48 h to 0 h. But it looks like the authors may be comparing 48 h to 24 h? Please double-check calculations and when stating % reduction, please calculate the reduction based on the control/check at 0 h.
Response 8: Thanks very much for reviewer's query. As mentioned by the reviewer, we did not emphasize that the comparisons at each time point were made with respect to the 0 h reference. Regarding this issue, we have provided supplementary explanations (line 125-128).
9. Line 119: What is meant by “any of these factors”. Does it only refer to what was noted about FQ118 in the text, or does it mean all the data given in Figure 2? Also, why only look for significance at “adjacent time points”? This is misleading because it suggests that there is no significance across the experiment for FQ119. For example, stem diameter of FQ118 at 18 h is significantly less than at 0 h (using the letters, which I assume indicate significant differences). In fact stem diameter for FQ119 at 48 h is significantly less than at 24 h, which would be “adjacent time points.”
Response 9: Thanks to the reviewer's valuable suggestions. Here, "any of these factors" refers to all the data presented in Figure 2. In fact, what we aim to show is the differences among various growth indicators between FQ119 and FQ118. This is indeed the case. Within 48 hours after drought treatment, the changes in each time point among various indicators of FQ119 were not significant. This also reflects that under drought stress, FQ119 tomato can maintain a good growth state, further demonstrating its strong drought tolerance. According to the reviewer's comments, we have made revisions to the parts that were ambiguous or unclear in description (line 125-128).
10. Lines 128-130: I agree that after 18 h, the trends for FQ119 seem somewhat “stable.” But the authors state that an “opposite trend was observed” for FQ118. However, looking at Figure 3, the data points for FQ118 look erratic and I don’t think the authors should be confident about any trends for FQ118, even after 18 h. For POD, isn’t there an increase from 18 h to 48 h (including 24 h), for FQ118, like (but not as clear as) FQ119?
Response 10: Thanks to the reviewer's valuable suggestions. Indeed, as the reviewer pointed out, the description of the regularities here is very difficult because the changes in the activities of various enzymes measured after the drought treatment of FQ118 are chaotic and disorderly. In response to the reviewer's requests, we have made some revisions to the descriptions and endeavored to highlight the key points of the changing trends as much as possible (line 138-147).
11. Figure 3: The caption for Figure 3 indicates that standard error is shown, but it is not shown in the figure. I think the standard error bars should be added.
Response 11: Thanks to the reviewer's valuable suggestions. The determination of each indicator in Figure 3 of this study actually has error bars, but the differences in the values between repetitions are not significant, and the circle shape of the values is too large, making the error bars hardly visible. We are also grateful for the reminder from reviewer. Therefore, we have reorganized and redrawn this figure (line 152), and provided the relevant raw data in the attachment. We hope this can improve the efficiency of interpreting our results in this part.
12. Lines 228-240: When the authors refer to genes that are “up-regulated” or “down-regulated” it is not always clear if they mean up/down in FQ118 or FQ119. Please clarify.
Response 12: Thanks very much for reviewer's query. The analysis we conducted here pertains to the comparison of DEGs between groups (A_vs_F). Here, ‘A’ stands for FQ118 and ‘F’ for FQ119. This point has been mentioned at line 158-162. Moreover, the screening of DEGs must be carried out for those genes that are enriched in both materials. Therefore, there is no such definition as whether it is the gene of FQ118 or FQ119.
13. Line 330 (and/or line 490): I assume that for pTRV2-PDS, the “PDS” means RNA interference for phytoene desaturase, which would be expressed as bleaching, and this is used as a marker? Please clarify for readers who aren’t familiar.
Response 13: Thanks to the reviewer's valuable suggestions. We have provided a brief explanation and appropriate clarification regarding the mechanism of PDS as an indicator (line 530-533).
14. Line 471: Were seeds sprouted in a “dish” and then transplanted into a pot? Or should the word “dish” be changed to “pot”?
Response 14: According to the reviewer's suggestion, we have change the word “dish” into “pot” (line 500).
15. Lines 476-477: How was PEG “added to the roots”?
Response 15: Thanks very much for reviewer's query. We have added the methods description of PEG treatments to the tomato roots at line 504-510.
Thanks again to the reviewer for taking time out of their busy schedules to review our manuscript. As suggested, we have developed the quality of English language throughout the whole manuscript. We hope that the above changes can improve the level of our manuscript to meet the standards of journal publication.
Round 2
Reviewer 2 Report
Comments and Suggestions for Authors
The manuscript is much improved. There are just a few improvements to suggest:
Lines 19 and 24 (and maybe elsewhere, please check): “below-ground” was changed to “down-ground” but this should have stayed “below-ground”
Lines 96-97: The observation in parentheses should either be cited, or noted as “data not shown” or “unpublished observations” or something similar. Please do the same for lines 113-115.
Lines 155-156: Don’t start a sentence with “And”. Also, as written, the letters not only designate significant differences among times within a line, but also between lines. I don’t think this is what is intended. I recommend changing to “…significant differences among time periods within a line…” or something similar.
Figures 10 and 11: I didn’t catch this in the original review, but I don’t fully understand the y-axis label “Ratio of variance expression (FC)”. Using the word “variance” makes it seem like the figures are showing the ratio of stastical variances, but I think it’s the ratio of expression levels. If I’m wrong, please ignore. Why not just “Relative expression level (FC)” as in Figure 12a? Also, in the captions, please define “FC” as “fold-change” (if that is correct).
Lines 504-511: The description of PEG treatment is written as a protocol directing someone to do this in the future, rather than what was already done in this research, which is awkward. For example, it should be written to indicate that the PEG solution was prepared and the control group was watered the same way, in other words, past tense, describing something that was already done. Please modify.
Comments on the Quality of English LanguageThe manuscript's English can still be improved. I assume the editors will assist.
Author Response
Dear Editor and Reviewer 2:
Thank you for your letter and for the reviewer’s comments concerning our manuscript entitled “Screening and Identification on Drought-Tolerant Genes in Tomato (Solanum lycopersicum L.) Based on RNA-seq Analysis”. (ID: plants-3596571). Those comments are all valuable and very helpful for revising and improving our paper, as well as the important guiding significance to our researches. We have studied comments carefully and have made correction which we hope meet with approval. Revised portion are marked in the paper. The main corrections in the paper and the responds to the reviewer's comments are as flowing:
1. Lines 19 and 24 (and maybe elsewhere, please check): “below-ground” was changed to “down-ground” but this should have stayed “below-ground”
Response 1: Thanks very much for reviewer’s valuable suggestion. We have made amendments to the inappropriate descriptions (lines 19, 104, 123 and 552). And we have re-edited the Y-axis information in Figure 2 (line 130). Please check it.
2. Lines 96-97: The observation in parentheses should either be cited, or noted as “data not shown” or “unpublished observations” or something similar. Please do the same for lines 113-115.
Response 2: Thank you for the comments from the reviewer. We have placed the description of this part (original lines 96-97) in the ‘Plant Materials and Growth’ section (lines 504-505). And original lines 113 to 115 describe precisely the changing trends of the plant height and stem diameter of the tested tomato plants as shown in Figure 2 (line 130).
3. Lines 155-156: Don’t start a sentence with “And”. Also, as written, the letters not only designate significant differences among times within a line, but also between lines. I don’t think this is what is intended. I recommend changing to “…significant differences among time periods within a line…” or something similar.
Response 3: Thanks very much for reviewer’s valuable suggestions. According to the requirements, we have modified the annotations of the significant marking letters on the Figure 2 and 3 (lines 132-133 and line 154-155).
4. Figures 10 and 11: I didn’t catch this in the original review, but I don’t fully understand the y-axis label “Ratio of variance expression (FC)”. Using the word “variance” makes it seem like the figures are showing the ratio of stastical variances, but I think it’s the ratio of expression levels. If I’m wrong, please ignore. Why not just “Relative expression level (FC)” as in Figure 12a? Also, in the captions, please define “FC” as “fold-change” (if that is correct).
Response 4: Thanks very much for reviewer’s valuable suggestions. Indeed, our description of the biological definition of the Y-axis is incorrect. In fact, what was measured was the relative expression level (FC). We have already modified Figures 10, 11 and 12 (a) (lines 309-311). Subsequently, we added descriptive supplements regarding the biological meanings represented by the horizontal and vertical axes in the figure notes (lines 313-315, lines 344-345 and lines 363-364).
5. Lines 504-511: The description of PEG treatment is written as a protocol directing someone to do this in the future, rather than what was already done in this research, which is awkward. For example, it should be written to indicate that the PEG solution was prepared and the control group was watered the same way, in other words, past tense, describing something that was already done. Please modify.
Response 5: Thank you for the comments from the reviewer. According to the requirements, we have made modifications to the method here. Please check it (lines 507-512).
Thanks again to the reviewer for taking time out of their busy schedules to review our manuscript. We hope that the above changes can improve the level of our manuscript to meet the standards of journal publication.